# Motives and Barriers Related to Physical Activity within Different Types of Built Environments: Implications for Health Promotion

**DOI:** 10.3390/ijerph19159000

**Published:** 2022-07-24

**Authors:** Marlene Rosager Lund Pedersen, Thomas Viskum Gjelstrup Bredahl, Karsten Elmose-Østerlund, Anne Faber Hansen

**Affiliations:** 1Department of Sports Science and Clinical Biomechanics, Faculty of Health Sciences, University of Southern Denmark, Campusvej 55, 5230 Odense, Denmark; tbredahl@health.sdu.dk (T.V.G.B.); kosterlund@health.sdu.dk (K.E.-Ø.); 2Department of Research and Analysis, University Library of Southern Denmark, 5230 Odense, Denmark; annefaber@bib.sdu.dk

**Keywords:** infrastructure, walkability, cyclist infrastructure, neighbourhood parks, open spaces, sports facilities, motives, barriers, literature review, scoping review

## Abstract

Studies have identified individuals’ motives and barriers as main predictors of physical-activity behaviour, while other studies found physical-activity behaviour to be related to characteristics of the built environment. However, studies that have a combined focus on motives and barriers and the built environment are less common. This scoping review aims to provide knowledge about motives and barriers related to physical activity within different types of built environments to mitigate this knowledge gap. A systematic literature search was performed in four scientific databases and yielded 2734 articles, of which 31 articles met the inclusion criteria. The review identified four types of built environments within which motives and barriers were studied, including walkability, cyclist infrastructure, neighbourhood parks and open spaces and sports facilities. Several common motives recur across all four types of built environments, especially easy accessibility and good facility conditions. Conversely, poor accessibility and inadequate facility conditions are common barriers. Our review also showed how some motives and barriers seem to be more context-specific because they were only identified within a few types of built environments. This knowledge may help target future health-promotion initiatives in relation to urban planning and the importance of the environment on physical activity.

## 1. Introduction

Studies have shown that a lack of adequate physical activity is a societal health risk problem [1]. An inadequate level of physical activity among the general population increases the risk of obesity, several lifestyle diseases and decreased lifetime expectancy [2]. Thus, physical inactivity poses great individual and societal threat [1]. On the other hand, studies also show strong evidence for a positive effect of leisure-time physical activity on various physical [3,4,5] and mental health issues [6,7,8]. On that account, strong evidence for physical activity as an instrument for improving population health exists.

Based on systemic research, e.g., by Bronfenbrenner [9] and Sallis [10,11], and in relation to the Theoretical Domains Framework [12], it is well established that individual physical-activity behaviour is based upon a complex interaction, e.g., between intrapersonal, interpersonal, organisational, community and policy domains [12,13]. The literature argues that physical-activity behaviour is a complex matter and cannot be fully understood or promoted unless the complexity of these domains is considered when designing health-promotion programmes [10,14].

It is well documented that individuals’ motives and barriers are among the main predictors of physical-activity behaviour and sports participation [15,16,17,18]. Moreover, studies show that motives and barriers to physical activity are influenced by factors of socioeconomic status (SES), such as education and income influenced [19,20]. Furthermore, studies have found physical-activity behaviour to be associated with characteristics of the built environment [21,22].

However, studies that have a combined focus on motives and barriers as well as the built environment seem to be less commonly found. To mitigate this knowledge gap, this scoping literature review aims to provide knowledge about motives and barriers related to physical activity within different types of built environments. Therefore, the research aims to provide the motives and barriers for adults related to physical activity within different types of built environments. Such knowledge can inform health-promotion initiatives, which could, in turn, allow for more effective efforts when designing, building or creating built environments for physical activity.

## 2. Materials and Methods

### 2.1. Terminology of the Main Terms (Physical Activity, Motives, Barriers, and Built Environment)

Physical activity may be defined as ‘any bodily movement produced by skeletal muscle that results in energy expenditure’ [23], whereas sports activities involve elements of physical exertion and competition [24]. Physical activity is a broad concept that includes sports activities and physical activities, such as active transport and leisure-time activities in public areas or indoor facilities.

Motivation may be defined as a hypothetical construct used to describe the internal and/or external drive for the ‘initiation, direction, intensity and persistence of behaviour’ [25]. While motivation is used to describe the process when a person decides to take a specific action, the term ‘motive’ describes the specific rationale for performing a particular action [26]. For practical reasons, the term “motive” in this study covers both terms. The term ‘barrier’ describes the specific factor that hinders or inhibits a particular action [27].

The built environment is, in this study, defined as ‘the physical makeup of where we live, learn, work, and play—our homes, schools, businesses, streets and sidewalks, open spaces, and transportation options. The built environment can influence overall community health and individual behaviours such as physical activity and healthy eating’ [28]. Within this, we include aspects that are the most relevant to physical-activity behaviour, such as community design, public transport, the built environment for active transportation (walking and biking), pedestrian safety and other types of built environments in the local area, such as green areas, parks, open spaces, aesthetics and pleasantness, recreational facilities and sports facilities.

### 2.2. Identifying Relevant Studies

This study was conducted as a scoping review, as this review type permits an open and broad research question and thereby allows for a wide variety of study concepts [29]. The scoping review approach of Peters et al. [30] was followed. Preliminary searches on the central foci of the research question (motives, barriers, physical activity and built environment) were performed in several databases, and those with the most relevant search results were chosen for the final search. The final search strategies in the four databases were iteratively developed between two researchers and two research librarians. After completing the final search, the characteristics of identified studies, including study population, study design, variables and key findings, were mapped. A formal synthesis of evidence was not undertaken.

The systematic literature search was performed in the databases Global Health, Scopus, Sociological Abstracts and SPORTDiscus on 1st November 2019. It was updated on 20th September 2021 to include studies published between 2019 and 2021.

Studies with full text available in English, German, Danish, Swedish or Norwegian were included. There was no restriction on publication year. Duplicates were removed before review.

### 2.3. Search Strategy

The search strategy performed has been described in detail elsewhere [20]. The search strategy was a mix of “free text words” (searched in title, abstract and keywords) and “defined keywords” (chosen from the Thesaurus list of the databases, except for Scopus, which has no Thesaurus). The entire search strategy is listed in Table 1.

### 2.4. Inclusion and Exclusion Criteria

Articles were included if they met three criteria: (a) focus on physical activities in a broad understanding (e.g., walking, jogging, etc.); (b) focus on motives and barriers in a broad understanding and (c) focus on the built environment. As we were interested in contributing to a knowledge base with relevance to the Nordic countries, studies from Europe, Oceania and North America were included in order to enable transferability to Western culture and norms.

Articles were excluded if physical activities were aimed at specialised activities (e.g., hang gliding or parkour) or if the study focused specifically on people with disabilities, people with ethnic minority backgrounds or people under 15 years.

### 2.5. Study Selection

A total of 2734 references were identified in the four databases and imported into the library software Endnote. After the removal of 793 duplicates in Endnote, 1941 references were imported into the software Covidence, where 114 further duplicates were immediately removed. Thus, 1827 references underwent title and abstract review, of which 1717 did not meet inclusion criteria. Three of the four authors screened 50 articles together to validate the screening process internally. Hereafter, they screened a third of the remaining 1827 references each. In cases of disagreement, the reviewers achieved consensus. A total of 110 studies were assessed for eligibility, of which 79 were excluded for valid reasons. A total of 31 articles were included for further analysis and reviewed by two authors (Figure 1).

### 2.6. Data Extraction

For extraction of relevant data, a spreadsheet to present common themes from all included articles was developed. Data extraction included year of publication, study location, study design and method, study population, aim and variable categories, main results and conclusions (Appendix A). Parallelly with the data analysis, a two-step coding of each article (A&B) was performed into four types of built environments inspired by the categorisation of McCormack and Shiell [31], including A: (1) walkability, (2) cyclist infrastructure, (3) neighbourhood parks and open spaces and (4) sports facilities and B: (1) motives and (2) barriers, respectively.

## 3. Results

Overall, 31 studies dealt with motives and barriers related to physical activity within different types of built environments. Of these, 11 studies were from countries in Europe, followed by North America and Oceania with 9 and 3 studies, respectively. A total of eight studies included more than one country. The oldest studies included were published in 2002 and the latest in 2020. The most studies from any one country were from the USA (Table 2). The selected studies included both quantitative and qualitative studies (e.g., survey and interview studies), as seen in Appendix A.

We categorised the 31 included articles according to the type of built environment in relation to which motives and barriers were examined. Table 3 shows how many studies examined motives and barriers within each type of built environment. A few studies could be categorised into more than one category, so when summing up across all four types of built environments, the number adds up to more than 31.

According to the built environment in relation to physical activity, we identified 14 general motives, including accessibility, security, health benefits, performance, social support, well-being, social interaction, enjoyment, physical appearance, facility conditions, weather conditions, aesthetics, convenience and nature. Similarly, barriers according to the built environment in relation to physical activity can be categorised into nine general barriers, including difficult access, unsafety, traffic, weather conditions, facility conditions, time, economy, poor health, and personal factors. Below, we provide an overview of the categories of motives and barriers related to the four types of built environments for physical activity. At the end of the results section, a table overview of each type of built environment’s identified motives and barriers is presented.

### 3.1. Walkability

In general, the results show that use of the local area for walking depends on accessibility, facility conditions (pedestrian safety, aesthetics and pleasantness), social interaction and support [32,33,34,35]. The results showed a connection between facilities and walkability in the local area and the extent of walking in leisure time as well as walking as transport for adults and the elderly [16,36,37,38,39,40,41]. Accessibility and closeness to shops and other local destinations [36,42,43] seemed to be important for walking. One study showed that neighbourhood-built-environmental factors, e.g., shops (within a 1 km distance), were positively correlated with walking as transportation, even after adjusting for demographic and psychosocial factors [41]. Studies show that pedestrian safety can increase the elderly’s leisure time and transport walking [35,36,37]. Results also show that adults and the elderly who perceive the surroundings as more aesthetic walked more [16,43]. In addition, social support was perceived by the elderly as a motivating factor in connection with walking behaviour [38]. Among elderly citizens who walked, studies found a correlation between their perceived accessibility to facilities in the local area and their tendency to have their basic psychological needs met (‘autonomy’, ‘competence’ and ‘relatedness’) through walking [39,42,44]. The results indicate that the relationship between physical activity via walking and intrinsic motivation depends on how the elderly perceive the accessibility of facilities and the diversity of activities in the neighbourhood [42,43,44]. For middle-aged adults, the results show that having pedestrian access and feeling safe are essential factors for leisure-time walking and walking as transport [16,21,45,46]. For overweight adults, the results showed that accessibility to benches, facility conditions and aesthetics impacted whether they chose to walk in the local area [32].

Poor facility conditions, such as lack of streetlights, lack of pavements, lack of benches to rest, lack of accessibility of paths and other users of the facilities, were mentioned as barriers to outdoor walking in general [16,46].

### 3.2. Cyclist Infrastructure

Safety and cycle paths were found to be crucial for cycle behaviour. Furthermore, personal factors, such as health and well-being, were motives for cycling [47,48]. Other motives were, e.g., to improve and maintain health benefits and include physical activity in a busy everyday life [47,48]. One study showed that men were more likely to cycle in leisure time as transportation and over longer distances than women [47]. To a greater extent than men, however, women describe that fun and enjoyment, getting physical movement into a busy everyday life and getting fresh air were motivating factors. For women, social factors and facility conditions (e.g., aesthetic and maintenance status) were also crucial [48], and men were more likely to cycle for recreation and for transport, and they cycled longer than women [48]. Both men and women highlighted the social aspect of cycling as important [48]. For young people (15–20 years), short travel time, short distance, high autonomy (to be able to choose where and when), social interaction (primarily with friends), low costs, good access to different modes of transport and facilities (e.g., cycle paths) and suitable weather conditions (convenience) were essential conditions for their motivation for cycling as active transportation. Conditions such as security and health were not essential for cycling as active transport for young people, as was the case for adults [49].

For both women and men, barriers to cycling as transportation and cycling as recreation were facility conditions, such as traffic, pollution from cars, road rage and a lack of security [48]. Women reported more personal factors as barriers than men. In addition, to a higher degree than men, women emphasised distance to destinations and lack of access to a bicycle for transport as barriers [48]. In general, other barriers to cycling to work were perceived distance, travel time, becoming sweaty and stormy weather [47].

### 3.3. Neighbourhood Parks and Open Spaces

The results of ‘neighbourhood parks and open spaces’ show that access to nature can increase motivation for physical activity. This is primarily indicated by younger (16 to 18 years old) and the elderly people (79–94 years old) [43,50,51]. Perceived security is essential as motivation for spending time in nature for physical activity [50,51]. In addition, being physically active due to health-related goals was positively associated with increased use of, among other things, parks and public spaces [15]. Moreover, aesthetic conditions (how green an area is and how attractive it is assessed as being) were motives [16]. In addition, convenience, experiencing nature and the opportunity for social interaction were essential motives for outdoor activities [50,52]. Accessibility of outdoor green space areas, structures of facilities in the local community (e.g., design and useability) and diversity of physical facilities in the local environment were also crucial for physical activity [40,52].

Barriers to using nature for physical activity in the local area were lack of time and poor health [51]. In addition, traffic and lack of access to recreational areas were barriers [22,42,51]. When people perceived the local area as unsafe, it was a barrier to physical activity [51]. For the elderly (above 79 years of age), mainly, obstacles (e.g., uneven pavement) in the local area may make it difficult to get around, which was also a barrier to physical activity [53].

### 3.4. Sports Facilities

Most of the studies that examined motives and barriers for physical activity in sports facilities focused on gyms or sports clubs. As a result, gyms and sports clubs are central in the description of the findings below. The results show that the users of gyms and sports clubs varied in their characteristics, motives and goals [40]. Motives for participation in physical activity vary from setting to setting and according to gender [40]. Individuals primarily motivated by well-being, social interaction and performance typically preferred sports clubs over gyms. Men seemed to be more motivated to perform than women, while women seemed to be more motivated by fitness, health benefits and appearance for sports participation in sports clubs [54]. Individuals who were physically active in gyms and sports clubs seemed to be more motivated by physical health and social interaction than those who preferred outdoor activities in nature [50]. For older teenage girls, the results show that perceived easy accessibility and access to equipment (e.g., balls and bicycles in the home and parks in the local area) were related to an increased belief in maintaining regular participation in physical activity [55]. Proximity to affordable and well-maintained facilities was generally described as a motive for physical-activity behaviour [22].

Financial costs of organised physical activity in sport clubs or fitness club membership were mentioned as barriers [45,56]. Transport obstacles, e.g., having to drive some distance to participate and lack of transportation possibilities, were reported as barriers [45]. Other barriers were described as lack of accessibility to community programmes, lack of access to facilities—either in terms of distance, price or other accessibility issues—and poor maintenance [45,56,57,58].

An overview of motives and barriers to the possibilities for physical activity is presented in Table 4.

## 4. Discussion

This scoping review provides insight into the motives and barriers related to physical activity within different types of built environments, such as walkability, cyclist infrastructure, neighbourhood parks and open spaces and sports facilities. Several common motives recur across all four types of built environments, especially easy accessibility and good facility conditions. Conversely, poor accessibility and inadequate facility conditions are common barriers. However, our review also showed how some motives and barriers seem to be more context-specific in the sense that they are only identified as relevant within one or a few types of built environments. For example, some of the different motives and barriers across the four types of built environments are that a high degree of security is experienced as motivating, both regarding walkability and cyclist infrastructure, in neighbourhood parks and open spaces, whereas the usage of sports facilities for physical activity in gyms and sports clubs in the local area presents a particular barrier due to affordability.

### 4.1. Recommendations Based on the Review

It is well documented that individual motives and barriers are the main predictors of physical activity and sport participation [15,16,17,18]. Moreover, studies show that motives and barriers for physical activity are influenced by factors of SES, such as education and income [19,20]. Particularly in socially deprived areas, paying attention to good overview and lighting conditions is relevant. The paving quality of paths may initiate and maintain walking and cycling behaviour. Focus on safety in relation to traffic may be solved with areas and paths that are isolated from the rest of the traffic [16,46]. Finally, residential areas close to shops and benches may help weak, elderly or overweight people to participate in more outdoor physical activity.

The World Health Organization (WHO) defines health promotion as: ‘the process of enabling people to increase control over, and to improve, their health. To reach a state of complete physical, mental, and social well-being, an individual or group must be able to identify and to realize aspirations, to satisfy needs, and to change or cope with the environment’ [59]. This quotation emphasises the necessity of providing adequate built environments for physical activity among the population to avoid the rapidly increasing physical inactivity pandemic. The built environment for physical activity in the local areas is relevant and important for reaching this population. This review gives insight into some essential motives and barriers concerning physical activity possibilities. The knowledge presented here may be helpful for stakeholders when designing future health-promotion initiatives.

### 4.2. Strengths and Limitations

First, a strength of this review is that the analyses allowed us to present motives and barriers in relation to four types of built environments, including walkability, cyclist infrastructure, neighbourhood parks and open spaces and sports facilities and to point out specific motives and barriers for each type of built environment. Another strength is that the review is based on extensive and systematic literature searches in four databases with focus on both public health, physical performance and sociology, and thereby has provided broad insights into the literature in the field.

Concerning limitations, most of the included studies had a cross-sectional-study design. Furthermore, some of the reported findings are based on few studies. Finally, as this was a scoping review, a formal-evidence synthesis was not undertaken, which may be considered a limitation. However, the main goal of a scoping review is to identify the existing research and not necessarily to assess the quality of the results [30,60]. Therefore, reservations should be made in the interpretation of data.

Furthermore, this paper did not discuss issues on populations with special needs, as it has been discussed elsewhere [20].

## 5. Conclusions

This scoping review provided insight into the motives and barriers related to physical activity within different types of built environments, including walkability, cyclist infrastructure, neighbourhood parks and open spaces and sports facilities. Pronounced motives and barriers both across and within the four types of built environments were identified. Based on these results, it is recommendable that motives and barriers are differentiated according to the type of built environment. This knowledge may help target future health-promotion initiatives according to urban planning and the importance of the environment on physical activity.

## Figures and Tables

**Figure 1 ijerph-19-09000-f001:**
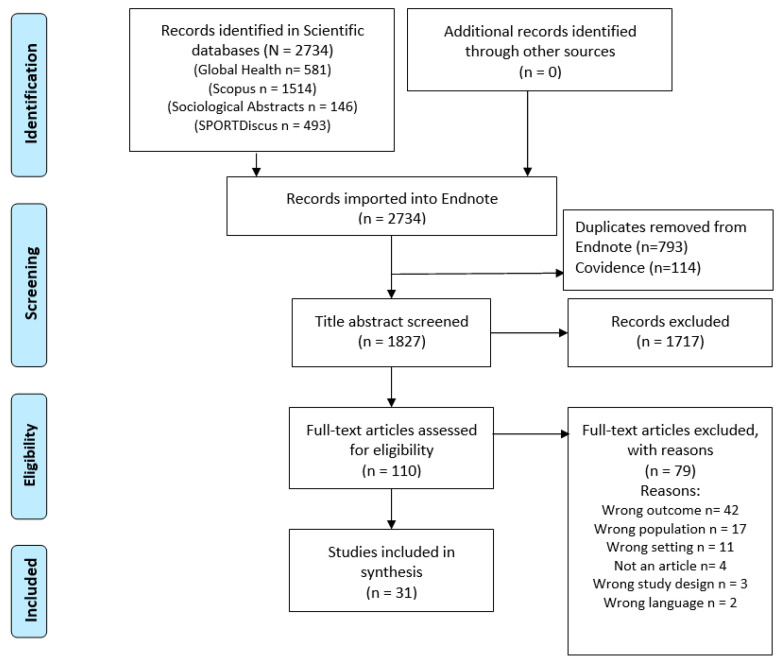
Flowchart of research results.

**Table 1 ijerph-19-09000-t001:** Search strategy. Search terms in the four selected databases.

	Search Blocks	Physical Activity	Motivation	Built Environment
Database				
**Scopus**		**Free text words:** Sport * W/2 participati * Physical W/2 exercis * Physical W/2 activit * active W/2 living active W/2 transportati * physical W/2 inactivity sedentary W/2 behavio? r * sedentary W/2 lifestyle * active W/2 lifestyle * recreational W/2 sport *	**Free text words:** Motivat * OR motive * Unmotivat * OR Demotivat * Amotivat * Barrier * OR constraint * Self W/2 determination W/2 theor * Social W/2 cognitive W/2 theor * Readiness W/2 change * “Stage * of change” Transtheoretical W/2 model * health W/2 action W/2 process W/2 approach motivational W/2 approach *	**Free text words:** Sport * W/2 (facility * OR hall * OR ground * OR arena *) Gym * Swimming W/2 pool * fitness W/2 (centre * OR center *) Leisure W/2 (centre * OR center *) Recreational W/2 (area * OR park * OR facility *) green W/2 space * sport * W/2 infrastructure * spatial W/2 accessibilit * built W/2 environment physical W/2 environment recreational W/2 facility * neighbo? Rhood * W/2 open W/2 space * (bicycle * OR walk *) W/2 path * (bicycle * OR walk *) W/2 trail * Bicycle * W/2 (lane* OR facility * OR track) Pedestrian * W/2 facilit * side W/2 walk * sport* W/2 club *
**SPORTDiscus**		**Defined keywords:** DE “SPORTS participation” DE “SEDENTARY behaviour” DE “SEDENTARY lifestyles” DE “SEDENTARY people”	**Defined keywords:** DE “MOTIVATION (psychology)” DE “TRANSTHEORETICAL model of change”	**Defined keywords:** DE “SWIMMING pools” DE “ACCESSIBLE design of parks” DE “ACCESSIBLE design of playgrounds” DE “ARCHITECTURE & recreation” DE “BICYCLE facilities” DE “TRAILS”
		**Free text words:** Same as Scopus Except for Proximity operator: “N2”	**Free text words:** Same as Scopus Except for Proximity operator: “N2”	**Free text words:** Same as Scopus Except for Proximity operator: “N2”
**Global Health**		**Defined keywords:** DE “sport” DE “physical activity” DE “active recreation”	**Defined keywords:** DE “motivation”	**Defined keywords:** DE “sports facilities” DE “sports centres” DE “sports grounds” DE “leisure centres” DE “recreational facilities”
		**Free text words:** Same as SPORTDiscus	**Free text words:** Same as SPORTDiscus	**Free text words:** Same as SPORTDiscus
**Sociological Abstracts**		**Defined keywords:** MAINSUBJECT.EXACT(“Sports Participation”)	**Defined keywords:** MAINSUBJECT.EXACT.EXPLODE(“Motivation”) MAINSUBJECT.EXACT.EXPLODE(“Constraints”)	**Defined keywords:** MAINSUBJECT.EXACT.EXPLODE(“Built Environment”) MAINSUBJECT.EXACT.EXPLODE(“Recreational Facilities”)
		**Free text words:** Same as Scopus Except for Proximity operator NEAR/2	**Free text words:** Same as Scopus Except for Proximity operator NEAR/2	**Free text words:** Same as Scopus Except for Proximity operator NEAR/2

**Table 2 ijerph-19-09000-t002:** Descriptive information about the included studies’ origins (continent and country).

Europe	No. of Studies
Belgium	1
Finland	2
Netherlands	2
Norway	3
Portugal	1
United Kingdom	2
**North America**	
Canada	1
USA	8
**Oceania**	
Australia	3
**More than one country** (3 of these studies contained only countries from Europe)	8

**Table 3 ijerph-19-09000-t003:** The number of studies that have examined motives and barriers within each of the four types of built environments.

Type of Built Environment	Studies
Walkability	16
Cyclist infrastructure	3
Neighbourhood parks and open spaces	10
Sports facilities	9

**Table 4 ijerph-19-09000-t004:** Motives and barriers related to physical activity within four different types of built environments.

	Motives	Barriers
**Walkability**	Accessibility Aesthetics Security Social support	Facility conditions Difficult access
**Cyclist infrastructure**	Accessibility Convenience Enjoyment Facility conditions (e.g., cycle paths) Health benefits Nature (e.g., get fresh air) Security Social interaction Well-being Weather conditions	Environmental factors Facility conditions Difficult access Personal factors Traffic Unsafety Weather conditions
**Neighbourhood parks and open spaces**	Aesthetics Convenience Facility conditions Health benefits Nature Security Social interaction	Difficult access Poor health Time Traffic
**Sports facilities**	Accessibility Facilities conditions Health benefits Performance Physical appearance Social interaction Well-being	Economy Facility conditions Difficult access

## Data Availability

All data has been published as Appendix A and was collected from published articles.

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
