# Peer review of "Motives and Barriers Related to Physical Activity within Different Types of Built Environments: Implications for Health Promotion"

_ijerph, 2022, doi:10.3390/ijerph19159000_

Round 1

Reviewer 1 Report

Dear Authors, your paper may be of interest for variety of readers. I have only some questions and comments. 

Why the Pubmed and Web of Science databases were not searched?

Minor remarks:

l.102 (Table 1) please explain which means: "free text word" and "defined keywords" and w/2. Genarally this table is not readable 

l. 198 - it's is good idea to state age- young people (15-20 yrs.)It is a pity that in other places the Authors do not specify the age (for example line 212/213- "younger and older people" The range of age is worth indicated because the term "older" is very broad. The same in line 225 for elderly - there are different  definition for elderly in different papers

Reviewer 2 Report

The article touches an interesting topic that merits further research, and is engaging and easy to read. You will find my detailed comments to the manuscript in the attached file.
